# Tuberculosis and isoniazid prophylaxis among adult HIV positive patients on ART in Northwest Ethiopia

Demeke Geremew[1]*, Habtamu Geremew[2], Mebratu Tamir[3], Mohammed Adem[4], Birhanemeskel Tegene[5], Biruk Bayleyegn[6]

1 Department of Medical Laboratory Sciences, Immunology and Molecular Biology Unit, College of Medicine and Health Sciences, Bahir Dar University, Bahir Dar, Ethiopia, 2 College of Health Science, Oda Bultum University, Chiro, Ethiopia, 3 Department of Medical Parasitology, School of Biomedical and Laboratory Sciences, University of Gondar, Gondar, Ethiopia, 4 Department of Immunology and Molecular Biology, School of Biomedical and Laboratory Sciences, University of Gondar, Gondar, Ethiopia, 5 Department of Medical Microbiology, School of Biomedical and Laboratory Sciences, College of Medicine and Health Sciences, University of Gondar, Gondar, Ethiopia, 6 Department of Hematology and Immunohematology, School of Biomedical and Laboratory Sciences, College of Medicine and Health Sciences, University of Gondar, Gondar, Ethiopia

* deme2112@gmail.com

**Data Availability Statement:** All relevant data are within the paper and its Supporting Information files.

## Abstract

### Background

Although antiretroviral therapy (ART) can avert tuberculosis (TB) incidence among human immunodeficiency virus (HIV) infected patients, the concomitant use of ART with isoniazid (INH) has a paramount effect. Despite this evidence, there is a paucity of data regarding TB incidence among HIV patients on ART with and without isoniazid prophylaxis and its predictors. Thus, this study sought to assess the incidence and predictors of TB among adult HIV positive patients on ART.

### Methods

This was a hospital based retrospective study including 368 adult HIV positive patients on ART in Gondar comprehensive specialized hospital between January 1, 2016, and April 30, 2019. Data was extracted from clinical laboratory and HIV care ART follow up clinic. The bivariable and multivariable regression models were used to ascertain predictors of incident TB. Data was analyzed using SPSS version 20 software.

### Results

A total of 335 adult HIV positive patients were included in the analysis, of whom, 56 (16.7%) were developed incident TB. Being ambulatory and bedridden (AOR: 2.2, 95% CI: 1.1, 4.6), advanced WHO clinical HIV disease stage (III and IV) (AOR: 3.2, 95% CI: 1.6, 6.1), not taking INH (AOR: 2.8, 95% CI: 1.3, 5.9), and baseline CD4$^+$ T cell count $\leq$ 200 cell/mm$^3$ (AOR: 3.6, 95% CI: 1.8, 7.2) were found to be the predictors of tuberculosis incidence.

**Funding:** The authors received no specific funding for this work.

**Competing interests:** The authors have declared that no competing interests exist.

**Abbreviations:** AOR, Adjusted odds ratio; ART, Antiretroviral therapy; CD4, Cluster of differentiation 4; CI, Confidence intervals; COR, Crude odds ratio; HIV, Human immunodeficiency virus; IPT, Isoniazid preventive therapy; INH, Isoniazid; IRB, Institutional review board; PLHW, People living with HIV; SD, Standard deviation; TB, Tuberculosis; UoGCSH, University of Gondar Comprehensive Specialized Hospital; WHO, World health Organization.

## Conclusion

The study indicated a high TB incidence among HIV positive patients in Gondar. Therefore, scaling up the isoniazid preventive therapy program and its strict compliance is necessary to avert HIV fueled tuberculosis in HIV endemic areas.

## Background

Globally, TB remained one of the main problems of health to mankind, especially in developing countries of sub-Saharan Africa, and is resurging again after the emergence of HIV pandemic [1]. Epidemiological studies indicated that HIV is a trigger for TB incidence, and people living with HIV (PLWH) have 20 times higher risk of developing TB related to those without HIV comorbidity [2]. This is partly due to HIV causes downregulation of the immune system directly by killing the host $CD4^+$ T cells [3].

Antiretroviral therapy has a significant effect in averting TB incidence in PLWH. However, HIV infected patients taking ART remained susceptible to TB, and TB is still the leading cause of mortality among HIV positive patients taking ART [4, 5]. As a result, world health organization (WHO) recommended isoniazid preventive therapy (IPT) to avert the development of TB among PLWH besides prompt ART initiation [4]. Recently established studies indicated that the concomitant use of IPT with ART have synergetic effect in preventing TB among HIV infected patients. Despite this promising evidence, its implementation in the targeted group is compromised due to different reasons [5].

The government of Ethiopia planned to reduce TB related mortality by 90% and incident TB by 80% in 2030 compared to 2015 levels [2]. To assess this ambitious plan, an updated evidence related to TB incidence and associated factors including isoniazid (INH) prophylaxis intervention measures has immense importance. Therefore, this retrospective study explored TB incidence and its predictors among HIV positive adults at University of Gondar comprehensive specialized hospital. The results gained from this study will have a positive effect in IPT implementation, inform policy makers and program developers working at different levels of TB controlling measures, and also health care experts working in TB control and prevention measures.

## Methods

### Study area, design, and period

A retrospective study was conducted in University of Gondar comprehensive specialized hospital (UoGCSH), from February to June 2019. The UoGCSH is one of the pioneer teaching hospitals in Ethiopia, and is located in Amhara region, Northwest Ethiopia. The hospital has eight different laboratory sections, including ART laboratory for $CD4^+$ T cell and viral load counts, which provides diagnostic, teaching, and research services for the university community, Gondar town inhabitants, and the nearby Woreda people at large.

### Study participants and eligibility

All adult HIV positive patients aged from 18 to 60 years who were newly starting ART at UoGCSH ART clinic and who had at least one follow up within 6 to 12 months from January 1, 2016 to June 1, 2019 were included in this study. However, all children aged under 18 and senior citizens aged above 60 were not included as their immunity to infection is altered, and hence it may have effect on baseline $CD4^+$ T cell counts that may lead to wrong interpretation

of the result. Similarly, those adults diagnosed as having TB at baseline during ART initiation and patients with incomplete baseline data for important variables such as functional status, WHO clinical stage, CD4$^+$ T cell count levels, and isoniazid prophylaxis status were excluded from the study.

## Sample size determination and sampling procedures

The sample size was calculated by using single population proportion formula through EPI INFO version 7.2.2.6 with the assumption of 95% level of confidence, 5% marginal error, and by taking 16.58% TB incidence among adult HIV positive patients from previous study in Ethiopia (Geremew *et al*., 2020) [1]. Given these assumptions, the sample size was estimated to be 217. Moreover, considering 10% expected incomplete data record the final sample size was 245. Nevertheless, to increase the power of the study we recruited a total of 368 study participants who didn't have baseline TB in this study. The records of all adult HIV positive patients initiating ART without baseline TB comorbidity and recorded from January 1, 2016 to June 1, 2019 were sorted from the electronic database to select the study participants. Thereafter, 368 study participants were selected using a simple random sampling technique through computer generated numbers.

## Data collection tool and procedures

Data abstraction tool was prepared from ART monitoring and evaluation forms. After preparing the extraction tool, consistency and completeness of the tool was verified compared to data recording systems by randomly selecting and completing a few chart reviews. Thereafter, minor amendments of the data collection tool were made accordingly before the beginning of data collection to ensure data quality. The tool includes sociodemographic, baseline clinical and INH prophylaxis associated variables. Data collectors were trained regarding each description of the tool and the way they collect data from patient charts. In order to pick charts from patient chart room, a medical record unique identifier number was first taken from the electronic record. Then, data was collected from the patient's medical charts using prepared data abstraction tool. A commonly agreed code by data collectors was used to avoid unintentional recollection of data. All selected patient charts fulfilling the eligibility criteria were reviewed and data abstraction was completed in this way.

## Measurements

The development of new TB within 12 months after ART initiation regardless of the time event was the dependent variable. Whereas the predictor variables were; sociodemographic characteristics (age, sex, and employment status), baseline clinical and laboratory characteristics (functional status, WHO clinical stage, and CD4$^+$ T cell counts), and other medications (isoniazid prophylaxis). This study mainly focuses on the number of new TB cases within a year rather than the time to new TB development for HIV positive adults after ART initiation till the end of the study. For this study, incidence of TB is defined as the occurrence of TB cases which is confirmed by bacteriological testes (at least one positive AFB microscopy, culture positive, Xpert MTB/Rif assay positive) or based on clinical decision of expert physician after initiation of ART during follow up [6].

## Data analysis

Data were entered into epidata and exported to statistical package for the social sciences (SPSS) version 20 software. Frequencies and percentages were used to summarize

characteristics of study participants. Bivariate logistic regression analysis was performed to determine the association of each individual independent variable with the outcome variable. As a result, crude odds ratio (COR) with corresponding 95% confidence intervals (95% CI) was obtained. Multiple logistic regression analysis was performed to assess simultaneously the association between multiple risk factors and the log odds of being positive for TB incidence. From this model, adjusted odds ratios (AOR) and 95% CI were obtained and then variables with P value <0.05 were considered as statically significant.

### Ethical considerations

Ethical clearance was obtained from the Institutional Review Board (IRB) of the University of Gondar, School of Biomedical and Laboratory Sciences. The IRB waived the requirement for informed consent to collect and retrieve data from the patients' medical chart. In addition, confidentiality was maintained during data extraction by using only unique identification codes rather than patient names.

## Results

### Baseline sociodemographic and clinical profiles of study participants

After exclusion of thirty-three (33) charts due to data incompleteness, 335 adult HIV positive patients' records were included based on the predefined inclusion criteria and considered in the final analysis.

From 335 participants included in the analysis, about 182 (54.3%) were females. The mean age of the study participants were 36.7 years with a standard deviation (SD) of 9.2 years, and 176 (52.5%) of the total subjects were in the age group of 31 to 45 years. The majority of the patients 201 (60%) were unemployed, 276 (82.4%) had working functional status, 215 (64.2%) had WHO clinical stage I and II, 119 (35.5%) had isoniazid prophylaxis, while 246 (73.4%) of the participants had CD4$^+$ T cell counts > 200 cell/mm$^3$ (Table 1).

### Tuberculosis incidence

The incidence of tuberculosis at the end of 12 months of follow up after ART initiation was 56/335 (16.7%). More specifically, TB incidence with and without INH prophylaxis was 8.4% and 21.3%, respectively. The incidence of TB among men and women were 27 (17.6%) and 29 (15.9%), correspondingly. Regarding WHO clinical HIV disease stage, the incidence of TB on HIV infected adults on stage I/II were 11.2% while among HIV patient who were on advanced disease stage (III or IV) the incidence was 26.7% (Table 2).

### Factors associated with incidence of TB among HIV infected adults

After adjusting the potential confounder using multivariant regression, variables including being ambulatory and bedridden functional status (AOR: 2.2, 95% CI: 1.1, 4.6), having advanced HIV disease stage III/IV (AOR: 3.2, 95% CI: 1.6, 6.1), not taking INH prophylaxis (AOR: 2.8, 95% CI:1.3, 5.9), and having low baseline CD4$^+$ T cell count (AOR: 3.6, 95% CI:1.8, 7.2) remained significantly associated with TB incidence among HIV infected adults (Table 3).

## Discussion

Tuberculosis remains one of the leading causes of morbidity, and is responsible for one third of death among people living with HIV globally, especially in developing countries including Ethiopia [7–9].

**Table 1. Baseline demographic and clinical profiles of study participants living with HIV in Gondar.**

| Variables | Frequency (n) | Percentage (%) |
|---|---|---|
| Age | | |
| 19–30 | 105 | 31.3 |
| 31–45 | 176 | 52.5 |
| 46–65 | 54 | 16.2 |
| Sex | | |
| Male | 153 | 45.7 |
| Female | 182 | 54.3 |
| Employment status | | |
| Employed | 134 | 40.0 |
| Unemployed | 201 | 60.0 |
| Functional status | | |
| Working | 276 | 82.4 |
| Ambulatory-bedridden | 59 | 17.6 |
| WHO stage | | |
| I/II | 215 | 64.2 |
| III/IV | 120 | 35.8 |
| INH prophylaxis | | |
| Yes | 119 | 35.5 |
| No | 216 | 64.5 |
| CD4$^+$ T cell count | | |
| > 200 cell/mm$^3$ | 246 | 73.4 |
| ≤ 200 cell/mm$^3$ | 89 | 26.6 |

**Table 2. TB incidence with different demographic and clinical profiles of study participants living with HIV in Gondar.**

| Variables | Total patients, (n = 335) | TB incidence | |
|---|---|---|---|
| | | Yes, n (%) | No, n (%) |
| Total patients, n (%) | 335 (100) | 56 (16.7%) | 279 (83.3) |
| Age years, Mean ± SD | 36.7 ± 9.2 | 38.2 (8.7) | 36.4 (9.3) |
| Sex, n (%) | | | |
| Male | 153 (45.7) | 27 (17.6) | 126 (82.4) |
| Female | 182 (54.3) | 29 (15.9) | 153 (84.1) |
| Employment status, n (%) | | | |
| Employed | 134 (40.0) | 23 (17.2) | 111 (82.8) |
| Unemployed | 201 (60.0) | 33 (16.4) | 168 (83.6) |
| Functional status, n (%) | | | |
| Working | 276 (82.4) | 39 (14.1) | 237 (85.9) |
| Ambulatory-bedridden | 59 (17.6) | 17 (28.8) | 42 (71.2) |
| WHO stage, n (%) | | | |
| I/II | 215 (64.2) | 24 (11.2) | 191 (88.8) |
| III/IV | 120 (35.8) | 32 (26.7) | 88 (73.3) |
| INH prophylaxis, n (%) | | | |
| Yes | 119 (35.5) | 10 (8.4) | 109 (91.6) |
| No | 216 (64.5) | 46 (21.3) | 170 (78.7) |
| CD4$^+$ T cell count, n (%) | | | |
| > 200 cell/mm$^3$ | 246 (73.4) | 32 (13.0) | 214 (87.0) |
| ≤ 200 cell/mm$^3$ | 89 (26.6) | 24 (27.0) | 65 (73.0) |

NB: SD = Standard Deviation INH = Isoniazid.

**Table 3. Predictors of TB incidence in adults living with HIV in Gondar.**

| Variables | TB incidence | | COR (95%CI) | AOR (95%CI) |
|---|---|---|---|---|
| | Yes, n (%) | No, n (%) | | |
| Age groups | | | | |
| 19–30 | 14 (25.0) | 91 (32.6) | 1 | 1 |
| 31–45 | 34 (60.7) | 142 (50.9) | 1.5 (0.7, 3.1) | 1.6 (0.8, 3.5) |
| 46–65 | 8 (14.3) | 46 (16.5) | 1.1 (0.4, 2.9) | 1.3 (0.5, 3.9) |
| Sex | | | | |
| Male | 27 (48.2) | 126 (45.2) | 1 | 1 |
| Female | 29 (51.8) | 153 (54.8) | 0.8 (0.5, 1.6) | 1.1 (0.6, 2.1) |
| Employment status | | | | |
| Employed | 23 (41.1) | 111 (39.8) | 1 | 1 |
| Unemployed | 33 (58.9) | 168 (60.2) | 0.9 (0.5, 1.7) | 0.9 (0.5, 1.7) |
| Functional status | | | | |
| Working | 39 (69.6) | 237 (84.9) | 1 | 1 |
| Ambulatory-bedridden | 17 (30.4) | 42 (15.1) | 2.3 (1.2, 4.4) | **2.2 (1.1, 4.6)** |
| WHO stage | | | | |
| I/II | 24 (42.9) | 191 (68.5) | 1 | 1 |
| III/IV | 32 (57.1) | 88 (31.5) | 2.9 (1.6, 5.2) | **3.2 (1.6, 6.1)** |
| INH prophylaxis | | | | |
| Yes | 10 (17.9) | 109 (39.1) | 1 | 1 |
| No | 46 (82.1) | 170 (60.9) | 2.9 (1.4, 6.1) | **2.8 (1.3, 5.9)** |
| CD4$^+$ T cell count | | | | |
| > 200 cell/mm$^3$ | 32 (57.1) | 214 (76.7) | 1 | 1 |
| ≤ 200 cell/mm$^3$ | 24 (42.9) | 65 (23.3) | 2.5 (1.4, 4. 5) | **3.6 (1.8, 7.2)** |

**NB:** COR: Crud Odd Ratio, AOR: Adjusted odd ratio. Bold numerals indicated that significantly associated with dependent variables.

In this study, the overall incidence of TB among ART initiated HIV positive adults was 16.7% (95% CI: 13, 21%) at the end of the follow up period. This is in line with study conducted in Debre Markos referral hospital, Northwest Ethiopia reported as 16.9% [2], in Shegaw Motta district hospital, Ethiopia 18% [10], in Hawassa University referral hospital, Southern Ethiopia 18.2% [11] and the finding in India 17% [12].

However, TB incidence rate revealed in this study is higher than previously established evidences in different regions of the country; 7.5% of TB prevalence in Gondar university hospital, Northwest Ethiopia [13], 10.3% overall pooled TB incidence in Ethiopia [5], 12.7% in Addis Ababa, Ethiopia [14], 7.2% in Arba Minch general hospital, Southern Ethiopia [15], 11.4% in Dessie referral hospital [16], 12.6% in Debre Markos referral hospital, Ethiopia [17] and 11% TB prevalence rate reported from Northwestern Tanzania [18]. This is probably attributed to the variation in sample size and/or in the duration of the follow up time. Partly, this could be because of the difference in TB case management and the use of tools to implementing the end TB strategy in different settings [19].

This study also revealed that HIV positive patients with bedridden and/or ambulatory functional status at enrolment were having 2.2 times higher risk of developing TB compared to their counterparts, working individuals (AOR: 2.2, 95% CI: 1.1, 4.6). This is similar with other previous studies conducted by Belew *et al.*, [20], Temesgen *et al.*, [2] and Alemu *et al.*, [14]. This might be due to bedridden or ambulatory HIV positive patients were unable to perform their day to day activities, as a result there might be a downregulated immune functions

including a decrease in their CD4$^+$ T cell counts and make them more susceptible to different viral and bacterial infections including TB [21, 22].

Advanced clinical WHO disease stage was another important independent predictor of TB incidence among HIV infected adults on ART in this study. Accordingly, the odds of developing TB was 3.2 times more likely for study participants with HIV disease stage III or IV as compared to patients on clinical early disease stage I and II (AOR: 3.2, 95% CI: 1.6, 6.1). This finding is in accordance with different earlier studies conducted nationally [2, 14, 23] and internationally [24]. This is partly due to patients with advanced clinical stage might be highly immunocompromised and subsequently leads to the recurrence of those low virulence opportunistic pathogens including TB infection [25]. However, a study conducted in Nigeria showed that there is no significant association between WHO clinical HIV stage with the incidence of TB among HIV infected people on ART [26].

In addition, this study demonstrated that taking INH prophylaxis was the significant independent predictors of TB among HIV infected adults on ART. Subsequently, study subjects who didn't took INH prophylaxis were having 2.8 times more risk of getting TB than those who took INH (AOR: 2.8, 95% CI:1.3, 5.9). This finding was consistent with earlier studies conducted in different study areas [17, 18, 27], suggesting that taking INH prophylaxis is the preventive factor for TB occurrence. This is partly due to the concomitant use of INH with ART have synergetic effect in preventing TB incidence among HIV infected patients [5] by improving the quantity and functional aspects of immunological markers including CD4$^+$ T cell counts.

Lastly, this study also showed that lower baseline CD4$^+$ T cell counts ($\leq$ 200 cell/mm$^3$) were another important determinant factor of TB incidence among HIV infected adults. Consequently, participants with baseline CD4$^+$ T cell counts $\leq$ 200 cell/mm$^3$ had 3.6 times increased risk of developing TB as compared to individuals who had CD4$^+$ T cell counts > 200 cell/mm$^3$. This finding is consistent with previous studies conducted in different parts of Ethiopia including Southern Ethiopia [28], Jimma, Ethiopia [23], and in another parts of the world such as Nigeria [26], India [12] and Tanzania [24]. This is partly because of low CD4$^+$ T cell counts in HIV infected patients indicates severe immunosuppression which makes the patients more susceptible to new or recurrent TB infection. Meanwhile, this finding is not consistent with other earlier finding where CD4$^+$ T cell counts was not significant predictors of TB among HIV infected patients receiving ART [29].

## Limitation of the study

The main limitation of this study was unable to detected virological failure which is the principal and direct indicators of immune status of the patients. The other limitation of the study was retrospective nature of the study design.

## Conclusion and recommendation

In this study, incidence of TB among HIV infected adults was significantly high. Adult patients on advanced HIV disease (stage III/IV), being ambulatory functional status, not taking INH and low baseline CD4$^+$ T cell counts were more likely to develop TB. Therefore, early detection and treatment of opportunistic infection should give a special attention. Furthermore, scaling up the INH prophylaxis program and its strict compliance is necessary to avert HIV fueled tuberculosis in the area.

## Supporting information

**S1 Dataset.**
(SAV)

## Acknowledgments

Our gratitude goes to the University of Gondar, study participants and health professionals working in ART clinic for their unreserved support during this study.

## Author Contributions

**Conceptualization:** Demeke Geremew.

**Data curation:** Demeke Geremew, Habtamu Geremew, Mebratu Tamir, Mohammed Adem, Birhanemeskel Tegene, Biruk Bayleyegn.

**Formal analysis:** Demeke Geremew, Habtamu Geremew, Birhanemeskel Tegene, Biruk Bayleyegn.

**Methodology:** Demeke Geremew, Mebratu Tamir, Mohammed Adem, Biruk Bayleyegn.

**Supervision:** Demeke Geremew, Mohammed Adem.

**Validation:** Demeke Geremew, Habtamu Geremew, Mebratu Tamir, Mohammed Adem, Birhanemeskel Tegene, Biruk Bayleyegn.

**Visualization:** Demeke Geremew.

**Writing – original draft:** Demeke Geremew, Habtamu Geremew, Biruk Bayleyegn.

**Writing – review & editing:** Demeke Geremew, Habtamu Geremew, Mebratu Tamir, Mohammed Adem, Birhanemeskel Tegene, Biruk Bayleyegn.

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
