## [Decision Letter · Decision Letter 0]

9 Feb 2022

PONE-D-21-18222Tuberculosis and isoniazid prophylaxis among adult HIV positive patients on ART in Northwest Ethiopia. a cohort study.PLOS ONE

Dear Dr. Geremew,

Thank you for submitting your manuscript to PLOS ONE. After careful consideration, we feel that it has merit but does not fully meet PLOS ONE’s publication criteria as it currently stands. Therefore, we invite you to submit a revised version of the manuscript that addresses the points raised during the review process.

 Although the manuscript is fairly well written, the study design is not robust and specially the objectives of the study. The conclusions are majorly based on presumptions and not on the robustness of data.

We look forward to receiving your revised manuscript.

Kind regards,

Sarman Singh, MD, FRSC, FRCP

Academic Editor

PLOS ONE

Journal Requirements:

2. Thank you for including your ethics statement: "Ethical clearance was obtained from the Institutional Review Board (IRB) of the University of Gondar, School of Biomedical and Laboratory Sciences. Besides, informed consent was obtained from the school and ART focal person to collect and retrieve the data from the patients’ chart. In addition, confidentiality was maintained during data extraction by using only unique identification codes rather than patient names.

a) Please provide additional details regarding participant consent. In the ethics statement in the Methods and online submission information, please ensure that you have specified (1) whether consent was informed and (2) what type you obtained (for instance, written or verbal, and if verbal, how it was documented and witnessed). If your study included minors, state whether you obtained consent from parents or guardians. If the need for consent was waived by the ethics committee, please include this information.

None

Additional Editor Comments:

Although the manuscript is fairly well written, I agree with the reviewers that study design is not robust and specially the objectives of the study. The issue regarding model of the study is raised by both reviewers and therefore, author need to demonstrate the aim of the study with pretested instrument.

Reviewers' comments:

Reviewer's Responses to Questions

**Comments to the Author**

1. Is the manuscript technically sound, and do the data support the conclusions?

Reviewer #1: Partly

Reviewer #2: Partly

2. Has the statistical analysis been performed appropriately and rigorously? 

Reviewer #1: Yes

Reviewer #2: No

3. Have the authors made all data underlying the findings in their manuscript fully available?

Reviewer #1: Yes

Reviewer #2: Yes

4. Is the manuscript presented in an intelligible fashion and written in standard English?

Reviewer #1: Yes

Reviewer #2: Yes

5. Review Comments to the Author

Reviewer #1: I must congratulate authors for writing on this important topic beautifully. I've only two issues related to the manuscript.

1. The Tuberculosis incidence definition is not mentioned in the manuscript. How the tuberculosis diagnosis was done? Is it a routine process that the cases of HIV (PLHIV) at the time of HIV diagnosis are screened for tuberculosis?

2. Since the study has not tested the relation between the ART and tuberculosis, the statement "In this study, incidence TB among HIV infected adults on ART were significantly high." is not applicable as a concluding statement.

Reviewer #2: Authors have attempted to describe the incidences of TB among HIV and further the strength of association with predictors .Although this is nearer to retrospective/chart review which is sometimes consider the major concern .In my humble opinion it may be overcome (and should not be a major concern) if investigators have strong theoretical construct before commencing the study .This does not seem the case in this study. The analysis is driven by the data and not by the theory which makes the inferences fragile , incidental and highly corelated to each other. For example WHO staging and functional status are highly corelated intuitively and clinically . Keeping these 2 variables simultaneously may not assign any predictive power to the model .

Moreover the model diagnostics ( R-square, -2LL ) are not shown by the authors .It is importance to understand the ratio of explained variance to unexplained variance (noise ) by the model . They are requested to prepare competitive models in increasing complexity before directly proceeding for data analysis by intuitively and logically selecting the variables and then they should select the most parsimonious yet on with explaining the maximum variance in the data set .

Another point which I would request to authors to think outcome -measurement as dynamic phenomenon and not as a static one. As of now they have taken the outcome measurement regardless of time event which actually beats the fundamental purpose of cohort study . They may apply the cox proportion regression hazard model as to understand the differential probability (odds) of developing tuberculosis at different point of time .This finding may give an indication to clinical vigilance by treating physician.

There are some syntax error in the sentence formations in data analysis section ( page-12 /2nd line) and result section (page-13/2nd para) which may be rewritten by authors for clarity purpose.

6. PLOS authors have the option to publish the peer review history of their article (what does this mean?). If published, this will include your full peer review and any attached files.

Reviewer #1: No

Reviewer #2: **Yes: **Ankur Joshi

---

## [Author Response · Author response to Decision Letter 0]

8 Mar 2022

Authors’ response letter, Manuscript ID - PONE-D-21-18222 

Thank you, the editorials, and reviewers for your valuable time and effort to share your experience and review our manuscript in detail. We the authors have addressed all comments and suggestions forwarded from the editors and reviewers. Our point-by-point detailed scientific justification and technical improvements follows the reviewer’s comments, and all page numbers refers to the cleaned revised manuscript file. 

Journal Requirements:

Authors’ response: Thank you for this useful suggestion. As per the journal’s guideline, we corrected and amended the the manuscript font size of the headings and subheadings, reference style and other Plos One requirement throughout the document. Please you may check the revised manuscript once again. 

2. Thank you for including your ethics statement: "Ethical clearance was obtained from the Institutional Review Board (IRB) of the University of Gondar, School of Biomedical and Laboratory Sciences. Besides, informed consent was obtained from the school and ART focal person to collect and retrieve the data from the patients’ chart. In addition, confidentiality was maintained during data extraction by using only unique identification codes rather than patient names. 

a) Please provide additional details regarding participant consent. In the ethics statement in the Methods and online submission information, please ensure that you have specified (1) whether consent was informed and (2) what type you obtained (for instance, written or verbal, and if verbal, how it was documented and witnessed). If your study included minors, state whether you obtained consent from parents or guardians. If the need for consent was waived by the ethics committee, please include this information.

Authors’ response: Thank you for critically looking at the ethical issue. As this study was retrospective in nature and the data were accessed by reviewing medical records without direct contact of study participants, the school of biomedical and laboratory science, University of Gondar IRB waived the requirement for any informed consent.

None

Authors’ response: Thank you the editors again for your suggestion. Regarding to funding, we addressed it in the revised manuscript (bottom of page 14) by saying “The authors received no specific funding for this work.”

Additional Editor Comments:

Although the manuscript is fairly well written, I agree with the reviewers that study design is not robust and specially the objectives of the study. The issue regarding model of the study is raised by both reviewers and therefore, author need to demonstrate the aim of the study with pretested instrument.

Authors’ response: Thank you the editors for your appreciation. Hence the main objectives of the study was to determine the incidence of TB among HIV infected patients under ART and the outcome variable was dichotomous in nature , we use both bivariable and multivariable logistic regression model and all the goodness of model fitness was assessed by Hosmer-Lemeshow Test due to that the sample size was satisfactory. 

Reviewers' comments:

Reviewer's Responses to Questions

Comments to the Author

1. Is the manuscript technically sound, and do the data support the conclusions?

Reviewer #1: Partly

Reviewer #2: Partly

Authors’ response: Thank you all the reviewers. The three years data collected from 2016-2019 indicated that the incidence of tuberculosis among adult HIV infected participants on ART was (16.7%) and more specifically, TB incidence with and without INH prophylaxis was 8.4% and 21.3%, respectively. Based on this data we concluded that the incidence of TB was high in the study area. This means that the data can support the conclusion and it is very informative indication for policymakers. 

2. Has the statistical analysis been performed appropriately and rigorously?

 Reviewer #1: Yes

Reviewer #2: No

Authors’ response: We are thankful to all reviewers for your great appreciation. To ensure the strength of the association of explanatory variables with the outcome variables, we perform bivariant and multivariant logistic regression which is relevant for the objective of this study. 

3. Have the authors made all data underlying the findings in their manuscript fully available?

 Reviewer #1: Yes

Reviewer #2: Yes

Authors’ response: Thank you for your appreciation.

4. Is the manuscript presented in an intelligible fashion and written in standard English?

 Reviewer #1: Yes

Reviewer #2: Yes

Authors’ response: Thank you once again for your encouragement. 

5. Review Comments to the Author

Comments from reviewer 1

 I must congratulate authors for writing on this important topic beautifully. I've only two issues related to the manuscript.

1. The Tuberculosis incidence definition is not mentioned in the manuscript. How the tuberculosis diagnosis was done? Is it a routine process that the cases of HIV (PLHIV) at the time of HIV diagnosis are screened for tuberculosis?

Authors’ response: This is a very interesting comment and we the authors also strongly agree with your insight. The definition of TB incidence was defined in the revised document (bottom of page 6, last sentence of under the heading “measurement”).

2. Since the study has not tested the relation between the ART and tuberculosis, the statement "In this study, incidence TB among HIV infected adults on ART were significantly high." is not applicable as a concluding statement.

Authors’ response: Great thanks for your constructive comments. Yes, you are correct that, we didn’t assess the association between ART and TB in this study even if the study participants were on ART. Then, as per the reviewer comments, we carefully revised the conclusion parts of the document. Please you may check the revised manuscript once again. 

Comments from reviewer 2

 Authors have attempted to describe the incidences of TB among HIV and further the strength of association with predictors. Although this is nearer to retrospective/chart review which is sometimes consider the major concern. In my humble opinion it may be overcome (and should not be a major concern) if investigators have strong theoretical construct before commencing the study. This does not seem the case in this study. The analysis is driven by the data and not by the theory which makes the inferences fragile, incidental and highly corelated to each other. For example, WHO staging and functional status are highly corelated intuitively and clinically. Keeping these 2 variables simultaneously may not assign any predictive power to the model.

Authors’ response: Thank you the reviewer for your great ideas. Before commencing the data collection, we have looked for the theoretical aspects of the association of TB incidence on HIV-infected individuals. The mechanisms of TB infection among HIV-infected individuals were discussed in detail. Furthermore, we rule out the major explanatory variables of incidence of TB like HIV disease stage and their functional status of the patients after reviewing recently published literature around the globe. We strongly agree with you that in the case of the advanced disease stage the patient becomes hospitalized which indicates that they may correlate with each other. In order to remove potential confounders like those, we analyse multiple logistic regression after doing bivariable logistic regression as the type of data was dichotomies.

Moreover, the model diagnostics (R-square, -2LL) are not shown by the authors. It is importance to understand the ratio of explained variance to unexplained variance (noise) by the model. They are requested to prepare competitive models in increasing complexity before directly proceeding for data analysis by intuitively and logically selecting the variables and then they should select the most parsimonious yet on with explaining the maximum variance in the data set.

Authors’ response: Great thanks for sharing your statistical experience that was untouched by the authors. However, in the current study “Hosmer and Lemeshow” analysis was performed for assessing model fitness as the sample size was high instead of using the -2 loglikelihood or -2LL. Additionally, the likelihood ratio test does not always perform well, especially when data are sparse. R-Squared is a statistical measure of fit that indicates how much variation of a dependent variable is explained by the independent variables in a regression model. However, this type of model is applied for linear regression where the data type of dependent and independent was continuous variable which is distinctively different from our data type. So please consider as the current data type is dichotomous which didn’t fulfil the linear regression assumption criteria.

Another point which I would request to authors to think outcome -measurement as dynamic phenomenon and not as a static one. As of now they have taken the outcome measurement regardless of time event which actually beats the fundamental purpose of cohort study. They may apply the cox proportion regression hazard model as to understand the differential probability (odds) of developing tuberculosis at different point of time. This finding may give an indication to clinical vigilance by treating physician.

Authors’ response: Thank you for your detailed reviewing the analysis part of the document and strong supporting idea. In this study we didn’t apply Cox proportional hazards models because these models are usually applied to prospective studies that have a follow-up period during which the occurrence of events is observed. Whereas our study mainly focuses on the number of new TB cases within a year rather than the time to new TB development for HIV positive adults after ART initiation and we haven’t any follow-up data then logistic regression models were applied for this cross-sectional study.

There are some syntax error in the sentence formations in data analysis section ( page-12 /2nd line) and result section (page-13/2nd para) which may be rewritten by authors for clarity purpose.

Authors’ response: Many thanks to the reviewer for this positive feedback, which we appreciate. We have now revised all unclear sentences in the data analysis and result section to ensure the sentences are clear and understandable to the PLOS One ID audience. Please see over the revised manuscript once again.

---

## [Editor Report · Decision Letter 1]

29 Mar 2022

Tuberculosis and isoniazid prophylaxis among adult HIV positive patients on ART in Northwest Ethiopia.

PONE-D-21-18222R1

Dear Dr. Geremew ,

We’re pleased to inform you that your manuscript has been judged scientifically suitable for publication and will be formally accepted for publication once it meets all outstanding technical requirements.

Kind regards,

Sarman Singh, MD, FRSC, FRCP

Academic Editor

PLOS ONE

---

## [Editor Report · Acceptance letter]

13 Apr 2022

PONE-D-21-18222R1 

Tuberculosis and isoniazid prophylaxis among adult HIV positive patients on ART in Northwest Ethiopia. 

Dear Dr. Geremew :

I'm pleased to inform you that your manuscript has been deemed suitable for publication in PLOS ONE. Congratulations! Your manuscript is now with our production department. 

Kind regards, 

on behalf of

Professor Sarman Singh 

Academic Editor

PLOS ONE